# A Comparative Study on Impact Resistance of Cylindrical Structures with Cushioning Energy Absorbing Rings under Double Impact Loading

**DOI:** 10.3390/ma17030595

**Published:** 2024-01-26

**Authors:** Bo Zhang, Shunshan Feng

**Affiliations:** State Key Laboratory of Explosion Science and Technology, Beijing Institute of Technology, Beijing 100081, China; 3120195186@bit.edu.cn

**Keywords:** cushioning energy absorbing rings, double impact loading, metallic cell structure, negative Poisson’s ratio, energy absorption

## Abstract

In this paper, the impact resistance of a cylindrical structure with a buffer ring and an energy-absorbing ring under double impact loads is studied. Based on ABAQUS 2023 finite element software, a simulation model of a buffer ring structure with three different sibs was established, and the specimens were subjected to double impact loading. The results show that the impact resistance of the structure decreases with the increase in curvature radius. The increase in the thickness of the panel can effectively reduce the deformation difference between the center point of the panel and the maximum displacement point. The buffer ring composed of cell structure with negative Poisson’s ratio effect has better shock resistance under explosion load, while the buffer ring with hexagonal cellular structure has excellent kinetic energy shock resistance.

## 1. Introduction

The sandwich structure, because of its high energy absorption density, can be used with a lighter protection structure to achieve the same protection effect, making it a popular protection device used in a variety of fields [1,2,3]. Sandwiches structure generally composed of a panel, core layer, and the back, the core layer unit as the main energy absorption buffer structure, has been widely studied. The multi-cell structure composed of metal cells has excellent dynamic mechanical properties in sandwich structures. When it is subjected to extreme loads such as explosion impact, it can gradually weaken the impact by collapsing the micro-structure locally, so it has become a research focus [4].

In practical applications, cylindrical structures are commonly found, such as the surface of the rocket shell or ammunition [5,6]. The curved structure has better structural strength and impact resistance when subjected to external impact [7]. Therefore, the composite structure combining the curved sandwich structure and the polycellular structure has a better protective effect than the general homogeneous structure [8,9,10].

Different from the traditional cellular materials used in the above studies, negative Poisson’s ratio structures with tensile behavior show enhanced properties in fracture toughness, shear modulus, and vibration absorption [11]. Therefore, in recent years, people began to study negative Poisson’s ratio structure. Li et al. studied the deformation and failure mechanism of shell and perforated square plate under explosion load [12]. Yuan et al. studied the medium-speed impact response and post-impact bending properties of hybrid sandwich structures [13]. MA Mohammed et al. studied the influence of different failure criteria based on quadratic stress function on low-speed impact response of thick filament wound glass/epoxy cylindrical structures [14]. Liang et al. studied the failure mode and impact resistance of polyurethane-coated metal cylinders under multi-field coupling loads [15]. Zhang et al. studied the dynamic response and damage behavior of mixed corrugated sandwich structures under high-speed hail-ice impact [16]. V. Acanfora et al. evaluated the impact resistance of polymer honeycomb core composite sandwich panels [17]. Du et al. analyzed the energy absorption and protection properties of hollow column thin-wall structures with different structural forms under impact loads [18]. Chen et al. compared the explosion resistance of honeycomb sandwich plates [19]. Qi et al. studied the impact and near-burst response of honeycomb sandwich panels with honeycomb cores by experimental and numerical methods [20]. Duc et al. also analyzed and studied the dynamic response of curved shells with thickened honeycomb cores under explosion load [21]. Lan et al. compared and analyzed the impact resistance of honeycomb sandwich board and aluminum foam sandwich board under explosion load, and indicated that sandwich laminates with negative Poisson’s ratio attribute have better explosion resistance, and pointed out that the stiffness of the backplane is an important factor affecting the impact resistance of negative Poisson’s ratio structure [22].

In the previous research results, the loads suffered by different protective structures are all explosion loads, and there are few studies on the combined loading of explosion loads and kinetic energy impact. With the deepening of the research, the single loading mode is not enough to fully reflect the protective function of the protective structure, and can not effectively adapt to the actual complex working environment. At the same time, for the comparison standard of the protective effect, the vast majority of studies use the internal deformation of the protective structure to determine the advantages and disadvantages of the protective effect. The protection structure is more to sacrifice the protection structure itself, so as to protect the important internal structure, in practical applications, some structures have the dual needs of protection and function; however, in some devices, such as high-speed train heads, rocket hoods, and other structures, in order to ensure its normal work, the shape needs to provide stable aerodynamic force, which has strict requirements on the shape of the structure. After the protective structure is loaded by explosion or kinetic energy impact, elastic deformation or limited plastic deformation occurs, and normal working requirements can be maintained after the loading is completed. Aircraft wings, cowling, and other structures need to maintain a sound aerodynamic shape, that is, after their own impact load, they can still normally complete the work function, which requires their own protection structure with good impact resistance.

In this paper, three kinds of cylindrical structures with cushioning energy absorption rings are designed on the basis of previous studies. The impact resistance of the three structures under the action of double impact loads is compared and analyzed, and the principle of cushioning energy absorption is analyzed. Combined with the curvature radius and thickness combination, the structure is further optimized, and the impact resistance of the composite buffer ring under the action of double impact loads is studied.

## 2. Problem Description

### 2.1. Geometry Description

Figure 1 is a schematic diagram of the curved sandwich structure studied in this paper. It can be seen from the diagram that the sandwich structure consists of a panel, three groups of independent core layers, and a backplane. In order to intuitively reflect the internal structure of the structure, the diagram adopts the 1/4 model. The geometric characteristics of the structure are determined by the radius of curvature R, axial length L, panel thickness T0, core thickness Tc, core interval d1, and core width d2. In practical applications, for structures such as kinetic energy munitions, the material strength of the internal structure is much higher than the material strength of the panel and core layer, so the deformation of the backplane will not be considered in the subsequent analysis, and therefore, its geometric parameters will not be considered. The geometric parameters of the structure in this paper are selected in the Table 1.

Figure 2 shows the boundary conditions and loading modes of the simulation analysis, which are the same as the structural diagram. The 1/4 model is adopted in the simulation analysis, and symmetric constraints are adopted at the boundaries. The explosion load is located outside the cylinder structure, 100 mm away from the outer surface of the panel, and the load equivalent is 25 gTNT. The loading of kinetic energy loads is carried out by a linear kinetic energy body, which is also applied to the outside of the cylinder. In practical application scenarios, explosive loads and kinetic loads are often generated at the same time, but the time difference between the effects on the surface of the protective structure is very large, usually several orders of magnitude difference. Therefore, in this paper, the load-loading process is divided into two steps. Figure 2a is the schematic diagram of explosion load simulation analysis, and Figure 2b is the schematic diagram of kinetic energy impact load.

### 2.2. Buffer Ring Structure Design

In order to better study the influence of the core layer on the protective structure, three kinds of core layers with different structures are designed to form the buffer energy absorption ring. They are the foam core layer, negative Poisson ratio core layer, and hexagonal honeycomb core layer. In order to better integrate the advantages of curved surface structure and metal cell structure, it is necessary to redesign the cell structure.

Figure 3 shows the structure diagram of the metal cell and buffer ring. The binding characteristics of the cell are characterized by four independent parameters, namely upper arc length L1, middle arc length L2, unit height H (H = 2 h), and shell unit wall thickness t. These four independent parameters can be adjusted by including angles θ1, θ2 curvature radii R1, R2. In this article, set the unit height H to 1 mm and set the aspect ratio L1/H = 1. Figure 3a,b are the cell structure diagrams of negative Poisson ratio and hexagonal honeycomb structure, respectively, and Figure 3c,d are the corresponding buffer ring structure modeling diagrams, respectively.

In this paper, three different buffer ring structures are set at the same relative density of 0.148. In order to ensure the same relative density of the three buffer ring structures, the method of controlling the thickness of the shell element is adopted in this paper. The width of the buffer ring structure is the same, and the total length of the side cell of the buffer ring structure is different by integrating the total length of the side cell of the different sibling structures. By adjusting the thickness of the shell element, the product of the three is the same. For the same cell material, that is, the total mass of the buffer ring structure, the relative density of the buffer ring structure is the same. Through calculation, the thickness range of shell elements of negative Poisson ratio buffer ring structure is 0.184–0.203 mm, and that of hexagonal honeycomb buffer ring structure is 0.241–0.271 mm.

## 3. Finite Element Modeling

### 3.1. Modeling of Geometric Models and Boundary Conditions

All numerical simulations were performed using ABAQUS 2023/Explicit finite element program. Due to the symmetry of the problem, in order to shorten the simulation time, only a quarter of the panel was modeled, and corresponding constraints were defined on the symmetric plane, as shown in Figure 2. ABAUQS shell element S4R (4-node, reduced integral shell element) was used for grid division of negative Poisoned son’s ratio honeycomb core and hexagonal honeycomb core, and the mesh size was 0.5 mm. The front panel, back panel, and foam core were meshed by solid unit C3D8R. The unit size was 1 mm in the axis direction and 0.5 mm in the radial direction. The grid study showed that a smaller grid had little effect on the improvement of calculation accuracy, but it would significantly increase the calculation time, so the mesh size was selected. It is assumed that the surface of the panel is perfectly combined with the surface of the buffer ring. In order to minimize the penetration of the slave surface into the master surface at the constraint locations and not allow the transfer of tensile stress across the interface, hard contact is adopted in the normal direction of the shell element, and the tangential behavior is set by penalty function, with the friction coefficient of 0.3.

Volumetric viscosity is used to introduce damping due to volumetric strain and is particularly necessary when studying higher-order performance in high-speed dynamic analysis. Volumetric viscosity is introduced only as a numerical effect, so the stress at the point of the material does not take into account the influence of volumetric viscous pressure. In the simulation, the linear bulk viscosity parameter was set to 0.06 and the quadratic bulk viscosity parameter to 1.2.

### 3.2. Material Model and Verification

#### 3.2.1. Material Model of Panel and Buffer Ring

Compared with the published research papers, Zhu et al. [8] experimentally studied the dynamic response of hexagonal honeycomb structure under explosion load, and the structure of the specimen was basically similar to the study in this paper, which could be used to verify the simulation model in this paper. Material damping is Rayleigh damping, which consists of two damping parameters. Mass proportional damping (α) is the proportional coefficient of the mass matrix, which is mainly used to eliminate low-order oscillations. Proportional damping of stiffness (β) is the proportional coefficient of the stiffness matrix, which is mainly used to eliminate higher-order oscillations. In this paper, the panel material is Al-2024 aluminum alloy, and the core layer structural material is Al-3104 aluminum alloy, whose material parameters are shown in Table 2.

#### 3.2.2. Material Model of Aluminum Foam

The *CRUSHABLE foam model and the CRUSHABLE foam hardening option in ABAQUS can better describe the material dynamic response of aluminum foam under impact loads. According to the ABAQUS user manual [23], the Deshpande and Fleck [24] models are adopted for aluminum foam materials, and the yield criterion can be defined as
(1)$$\Phi =\hat{\sigma }-\sigma_{y}\leq 0$$
where $\sigma_{y}$ is yield stress, equivalent stress $\hat{\sigma }$ as follows:(2)$${\hat{\sigma }}^{2}=\frac{1}{\left[1+{\left(\frac{\alpha }{3}\right)}^{2}\right]}\left[{\sigma_{e}}^{2}+\alpha^{2}\sigma_{m}^{2}\right]\;$$
where $\sigma_{e}$ is the normal equivalent stress, and $\sigma_{m}$ is the average stress. Parameter *α* which controls the shape of the yield surface is a function of shaping the Poisson ratio $\nu_{p}$, whose expression is:(3)$$\alpha^{2}=\frac{9\left(1-2\nu_{p}\right)}{2\left(1+\nu_{p}\right)}\;$$

For the aluminum foam material, $\nu_{p}=0$, $\alpha =\sqrt{\frac{9}{2}}$

Yield stress can be expressed as:(4)$$\sigma_{y}=\sigma_{p}+\gamma \frac{\hat{\varepsilon }}{\varepsilon_{D}}+\alpha_{2}\ln\left[\frac{1}{1-{\left(\frac{\hat{\varepsilon }}{\varepsilon_{D}}\right)}^{\beta }}\right]$$
where $\hat{\varepsilon }$ is the equivalent strain, $\sigma_{p}$, $E_{P}$, $\alpha_{2},\gamma ,\beta \;$ and $\varepsilon_{D}$ is the material parameter and the relationship between them and foam density is:(5)$${\left\{\begin{array}{l} \left(\sigma_{p},\alpha_{2},\gamma ,\frac{1}{\beta },E_{P}\right)=C_{0}+C_{1}\left(\frac{\rho_{f}}{\rho_{f0}}\right)\\ \varepsilon_{D}=-\ln\left(\frac{\rho_{f}}{\rho_{f0}}\right) \end{array}\right.}^{k}$$
where $\rho_{f}$ is the foam density and $\rho_{f0}$ is the base material density. The parameters of $C_{0}$, $C_{1}$ and *k* are shown in Table 3.

### 3.3. Double Impact Load Modeling

#### 3.3.1. Blast Load Modeling

In this study, the explosion load was generated by the comwep (conventional weapon effect program) empirical model. The conwep algorithm developed by Kingery and Bulmash [26] is considered by combining reflection pressure (normal incident) and incident pressure (lateral incidence). The formula for pressure calculation is as follows:(6)$$P_{load}=P_{reflected}\cdot \cos^{2}\theta +P_{incident}\cdot \left(1+\cos^{2}\theta -2\cos\theta \right)$$
where is θ the incidence angle, $P_{incident}$ is the incident pressure and $P_{reflected}$ is the reflected pressure.

The CONWEP model in Abaqus/Explicit takes two parameters TNT equivalent mass and contrast distance. Cylindrical TNT was used in the experiment, whereas the charge used in the CONWEP model was spherical in shape. In order to obtain the equivalent mass of CONWEP, Li [10] proposed a numerical calculation method. The commercial software Autodyn 2022 R1 [26] was used to simulate the explosive load pressure exerted on the structure by the cylindrical TNT charge detonated in air isolation. In order to make the peak pressure of the spherical TNT charge imposed on the front panel the same as that of the cylindrical TNT charge, empirical Equation (7) was used to calculate the equivalent mass of the spherical TNT charge [27,28].
(7)$$\Delta p=\frac{6.1938}{Z}+\frac{0.3262}{Z^{2}}+\frac{2.1324}{Z^{3}}$$
where Δp(kp/cm^2^) is the peak overpressure, and Z(m/kg^1/3^) is the comparison distance, 0.3≤Z≤1.

In addition, Dharmasena [29] proved through experiments and numerical results that the cylindrical charge with the aspect ratio of central initiation was close to 1 (the charge size used in the experiment was: height = 15 mm, diameter = 15 mm), and the CONWEP model reasonably estimated the explosion wave profile. Therefore, the equivalent amount of CONWEP is 25 g and the action distance is 100 mm to carry out simulation research and experimental comparison.

#### 3.3.2. Kinetic Impact Load Modeling

In this study, the kinetic impact load is loaded by establishing a standard unit kinetic energy body and changing the predefined initial velocity to achieve different kinetic energy loading purposes. In this study, the deformation of the kinetic energy body is not considered, and it is set as a rigid body in the simulation. The geometric parameters of kinetic bodies are shown in Table 4. According to different kinetic energy parameters, the preset initial velocity is 4.47 m/s, 7.75 m/s, and 10 m/s, respectively.

### 3.4. Finite Element Model Verification

#### 3.4.1. Honeycomb Core Laminate Structure Verification

Since there is no dynamic response test on this new curved deformed honeycomb structure in the existing literature, the effectiveness of the numerical method is verified by the response test on the square honeycomb plate under the blasting load [8]. In this study, ACG-1/4-TK-5 specimens were selected and simulated in Abaqus/Explicit with the same boundary conditions, materials, and blasting loads as in Section 3.1, Section 3.2 and Section 3.3.

The results are shown in Figure 4. According to the deformation of the core layer, it can be divided into three areas: full folding area, partial folding area, and non-folding area. It is clear that the deformation/failure modes of honeycomb sandwich structures predicted by numerical and experimental results are almost identical. As shown in Figure 5, the experimental and numerical predictions of plate deflection after experimental and numerical predictions are compared. The numerical prediction is close to the experimental results, and the error is within the acceptable range. Correlation studies show that simulation models calibrated with the same material model, boundary conditions, mesh, contact conditions, and loading modes can be used for subsequent simulation studies.

#### 3.4.2. Verification of Aluminum Foam Core Structure

In this paper, the explosion response test of aluminum foam laminates conducted by Jing et al. [9] was simulated. The test specimens were composed of two identical aluminum alloy panels and an aluminum foam sandwich, which were completely fixed, the loading area was sxl = 250 × 250 mm, and the detonation distance was 100 mm.

Figure 6 shows the comparison between the test results and the finite element simulation results. The data points are all attached to the matching line, indicating that the simulation results are in good agreement with the experimental results. The deformation mode of the specimen is shown in Figure 7. It can be seen that the deformation mode of the simulation model is in good agreement with the test results, which proves that the numerical simulation results are effective.

## 4. Results and Discussion

### 4.1. Comparison of Deformation Modes

As shown in the attached table, specimens with G1_1_F represent curvature radius of 40 mm, thickness of front panel To = 2 mm, and thickness of core layer Tc = 5 mm. Specimens with foam core layer structure under explosion load have the same parameters as those with auxetic core layer (AUX) and Hexagon core layer (HEX), respectively. The expression of A/H/F in the table indicates that other structures and load parameters of the three buffer ring structures in this row are the same. The structural parameters and load types of specimens in each group can be seen in the table. The “center point” in the table is the midpoint of the upper surface of the panel at the section.

The deformation of three cylindrical structures is shown in Figure 8. Under the effect of the explosion load, the center point and the maximum deformation point shift process of the three specimen panels are shown in Figure 9, with the maximum displacement point in the middle of the interlayer interval. Under the action of explosion load, the deformation process of the center point and the maximum deformation point of the panel are basically the same. The process of deformation can be divided into the stage of loading, the bounce stage, and the oscillation stage. The loading phase is short, the explosion load is loaded in a very short time, and the test specimen reaches the maximum deformation. At the rebound stage, the three core layers of the core layer were all larger, and the amplitude of the core layer was especially significant. During the oscillating stage, the panel continuously oscillates for a certain amount of time, and the test of the foam core layer is significantly increased than the other core layers. The reason for this phenomenon is that the deformation of the specimen with foam core is significantly increased, which can lead to a significant increase in the time of the panel concussion.

Under different kinetic energy impacts, the displacement process of the center point and maximum deformation point of the three specimens’ panels is shown in Figure 10 and Figure 11. Under kinetic impact load, the deformation process of the center point and the maximum deformation point of the panel are similar, but the variation amplitude is obviously different. Compared with the three loading processes of explosion load, the action time of kinetic impact load is also very small, but the time of rebound and shock stage is relatively short. The system stabilizes more quickly. Different from the center point of the panel, the springback amplitude of the maximum deformation point increases significantly or even exceeds the original position. With the increase in load kinetic energy, the springback amplitude increases.

By comparison with Figure 10 and Figure 11, under different kinetic impact conditions, when a low-speed impact load is loaded, the dynamic response of the target structure is crucial, and the whole structure can respond to the impact, so more energy is absorbed by elasticity [30]. The buffer ring structure with negative Poisson’s ratio core layer and foam core layer has a similar dynamic response of the whole structure, so the change amplitude of the center point of the two panels is similar. For the maximum deformation point, in the early explosive loading process, the displacement gap between the center point and the maximum deformation point of the buffer ring structure with foam core layer is small. Therefore, in the subsequent kinetic energy loading process, the buffer ring structure continues to compress, and the displacement of the maximum deformation point continues to increase. For specimens with negative Poisson’s ratio buffer ring structure and honeycomb core buffer ring structure, the displacement gap between the center point and the maximum displacement point is large after explosion loading. The panel tends to be flat under kinetic impact loading, and the overall deformation of the maximum displacement point is reduced compared with before, so the deformation amplitude of the two is similar.

As shown in Figure 12, the deformation modes of the core layers of the three different specimens under load impact are significantly different. When subjected to impact loading, the AUX core layer with negative Poisson’s ratio effect can pull more cell structure into the impact reaction zone, and the cell structure will collapse and deform, absorb the impact energy, and reduce the molding deformation of the panel.

Kinetic energy load and explosion impact load have obvious differences in loading form, explosion load will put the material in a state of high strain rate, resulting in cell structure deformation. However, under kinetic energy impact loading, the proportion of elastic deformation in cell structure deformation will increase. The whole structure can react to shocks, so the structure undergoes greater elastic deformation, producing more oscillations, and dissipating energy. Specimens with HEX structure, as shown in Figure 10 and Figure 11, exhibit better resistance to kinetic impact, and with the increase in impact load, the advantages over other core layers are more obvious.

### 4.2. Influence of Curvature Radius on Impact Resistance

As an important structural parameter of curved surface structure, the radius of curvature has a great influence on the failure mode and impact resistance of composite structures. This paper studied specimens with five kinds of curvature radii and designed 15 groups of comparison tests with curvature radii ranging from 40 mm to 80 mm. The relationship between the center point and the maximum displacement point of the three sandwich structures with respect to curvature radii is shown in Figure 13. The curvature radius has the same influence trend on the deformation of the three sandwich structures. With the increase in the curvature radius, the deformation of the specimen also increases. The existence of different curvature radius changes the reflection characteristic of the structure to the explosion shock wave, and the smaller curvature radius can change the reflection Angle of the shock wave on the panel, thus reducing the explosion impulse of the incoming structure.

The relationship between the deformation and curvature radius of the specimen under the dual shock load is shown in Figure 14. Under the combination of double shock load, the test specimen containing the core layer buffer ring is less than the other two sets of specimens because of its better anti-kinetic energy, and the advantage is more obvious as the kinetic energy load increases. In the radius of the study, the three core layers are more sensitive to the change of the curvature radius than the other two cores. The maximum displacement point and the center point of the surface have the same trend.

Figure 15 and Figure 16, respectively, show the deformation comparison of the three structures under the action of explosion load and double impact load. With the increase in curvature radius, the compression of the FOAM core layer will increase. The deformation degree of the buffer ring boundary of each structure is obviously greater than that of the center, which also results in the maximum displacement point being more affected by the curvature radius than the center point. Under the double impact load, the deformation gap between the center point and the maximum displacement point decreases with the increase in kinetic impact load, and this gap also decreases with the decrease in curvature radius.

The comparison of energy absorption of each part of the three structures under different curvature radii is shown in Figure 17, where C and F, respectively, represent the buffer ring and the front panel. The analysis shows that:With the increase in the curvature radius, the total energy of the panel suction can be increased, the proportion of the panel suction energy is increased in the total energy, and the foam core layer is especially obvious, which can also be explained that the specimen with the foam core layer has the maximum deformation, and the increase in the curvature radius is more obvious.In the case of the explosion load, the three types of buffer ring structure suction can be less affected by the radius of curvature, and the addition of plastic deformation is less, which is proved to be the deformation of the whole elastic deformation of the buffer ring structure, not the absorption properties.In the same way, the absorption can increase with the curvature radius, and the buffer ring with the aux core layer increases with the radius of the curvature, and the absorption of the curvature of the core decreases, and in the case of small kinetic energy loading, it is especially obvious in the condition of small kinetic energy loading, which indicates that the negative poisson structure has a better energy absorption effect under the condition of high strain rate and high energy density loading.

### 4.3. Influence of Panel Thickness

Groups 2, 4, 6, and 8 in Table 4 compare and analyze the influence of different panel and core thickness. Figure 18 shows the comparison of deformation conditions of three structures under different thickness combinations under the action of explosion load. It can be seen that with the increase in the thickness of the panel, the deformation mode of the buffer ring tends to be the whole deformation, the boundary is clear, and the local plastic deformation of the panel decreases, which is especially obvious in the specimen with FOAM core layer.

Figure 19 and Figure 20, respectively, show the relationship between specimen deformation and panel thickness under explosion load and double impact load. The results show that, under the action of explosion load, increasing the thickness of the panel can significantly reduce the deformation of the maximum displacement point. However, if the total thickness of the panel and the core layer remains unchanged and the thickness ratio of the two is changed, increasing the thickness of the panel will increase the displacement of the center point of the specimen with the AUX and HEX core layer buffer ring structure. The effect is reversed for specimens with FOAM core layer cushion ring structure. With the increase in the thickness of the panel, the difference between the deformation of the center point and the deformation of the maximum displacement point decreases significantly, or even is basically the same. This point also shows the same rule under the action of double impact load, and with the increase in kinetic impact load, the difference of the deformation also decreases.

In order to better illustrate the above laws, Figure 21 shows the comparison of energy absorption of different parts under different thickness combinations. C and F represent the buffer ring and the front panel, respectively.

Under the action of explosion load, the energy absorption of each part decreases with the increase in the thickness of the panel. This is because when the total thickness is unchanged, the thickness of the panel increases, the thickness of the buffer ring decreases, and the energy absorption of the buffer ring structure decreases. At the same time, the energy dissipation of the panel increases, the plastic deformation of the panel decreases, and the deformation difference between the center point and the maximum displacement point decreases.Under the action of kinetic impact load, with the increase in the thickness of the panel, the panel energy absorption of each specimen increases, while the thickness of the buffer ring decreases, and the buffer ring energy absorption decreases. This law is the same for the specimens of the three structures.Under the action of kinetic impact load, the energy absorption of all parts of the structure increases with the increase in kinetic impact load. Among them, the energy absorption of the buffer ring structure of the specimen with AUX core layer structure increases significantly, and the increased amplitude increases significantly with the decrease in the thickness of the panel and even the energy absorption of the buffer ring exceeds that of the panel. This once again proves that the AUX core layer with negative Poisson’s ratio effect can perform better energy absorption under high energy density conditions.

## 5. Study on Impact Resistance of Composite Buffer Ring Structure

In order to better study the impact resistance of the buffer ring structure, further analyze its influencing factors, and find the optimal solution at present, combined with the analysis results of the previous sections, as shown in Figure 22 the buffer ring of the two core layer structure is compounded and superimposed. The simulation analysis results are shown in Figure 23 and Figure 24.

For the composite buffer ring structure, the protection effect of the layered buffer ring structure is better than that of the complete structure. This is because when the stress wave propagates in the structure, due to the existence of the interface, part of the shock wave energy cancels each other through the transmission superposition and the reflected shock wave so that the shock wave energy is consumed and the external impact resistance is better.

It can be seen from the Figure that under the action of explosion load, the lower layer is the buffer ring structure of the HEX core layer and the upper layer is the AUX core layer, which has the minimum deformation at the center point and the maximum displacement point. This shows that the AUX structure can have a better energy absorption effect when the strength of the lower part is higher. In addition, the HEX upper layer structure can effectively reduce panel deformation during dual impact loading, due to better overall support performance, which allows for better impact resistance when subjected to kinetic impact.

In Figure 25, the composition of the energy absorption contrast diagram is given in different combinations, and the analysis can be obtained:Under the explosion load, the sample suction of the foam core layer can be much higher than the other core, the core layer is more consistent, and the upper and lower levels have almost the same amount of suction energy.The absorbability of the composite buffer ring is higher than the suction performance of the panel, and the latter is three and a half times. This is clearly different from the single-core layer buffer ring.When the upper core layer of the specimen is AUX cell structure, the energy absorption characteristics of the buffer ring are positively correlated with the structural strength of the cell structure of the lower core layer, which also explains why the combination of AUX core layer on the upper layer and HEX core layer on the lower layer has the smallest deformation.

## 6. Conclusions

In this paper, the impact response performance of cylindrical structures with buffer rings under double impact loading is compared and analyzed. Multiple sets of control data are set, and the following conclusions are obtained:The cylindrical structure with a buffer ring is created, three kinds of buffer ring core layer cells are designed, and the simulation model is established. The rationality of the simulation model is verified based on the test data, and the dynamic response of the three core layer structures under the action of double impact loads is simulated and analyzed.For the loading of a single explosion load, the buffer ring structure with a negative Poisson ratio can make more core structures collapse and absorb energy with its negative Poisson ratio effect so that it has better impact resistance and minimum panel deformation. Under the action of double impact loads, the overall impact resistance of the honeycomb core layer is better than that of other core layers, showing better kinetic energy impact resistance.The presence of different curvations changes the reflection characteristics of the structure to the explosion shock wave. With the increase in the curvature radius, the reflection Angle of the shock wave in the panel increases, thus increasing the incoming amount of explosion impulse, resulting in the decline of the impact resistance of the structure, and the impact resistance of the core layer will increase this influence.With the increase in the thickness of the panel, the gap between the impact resistance of each buffer ring will be narrowed, and the energy absorption ratio of the panel itself will increase, but the quality of the overall protection structure will increase significantly.In the study of composite structure, the upper layer is the AUX core layer and the lower layer is the HEX core layer buffer ring structure. Under the action of explosion load, when shock wave propagates between interfaces, the wave impedance of the lower core layer is greater than that of the upper core layer, so that more shock waves are reflected back to the upper core layer, which makes negative Poisson function better than the core layer and has the best performance. The HEX core buffer ring has the best performance under double impact load, and the panel deformation is minimal.In the follow-up research work, the influence of structures with different negative ratio effects on the impact resistance of specimens will continue to be explored, and the characteristic parameters of structures need to be further optimized to achieve the optimal scheme of lightweight and energy absorption of protective structures. Meanwhile, the research on the energy absorption mechanism of composite structures needs to be further enhanced.

## Figures and Tables

**Figure 1 materials-17-00595-f001:**
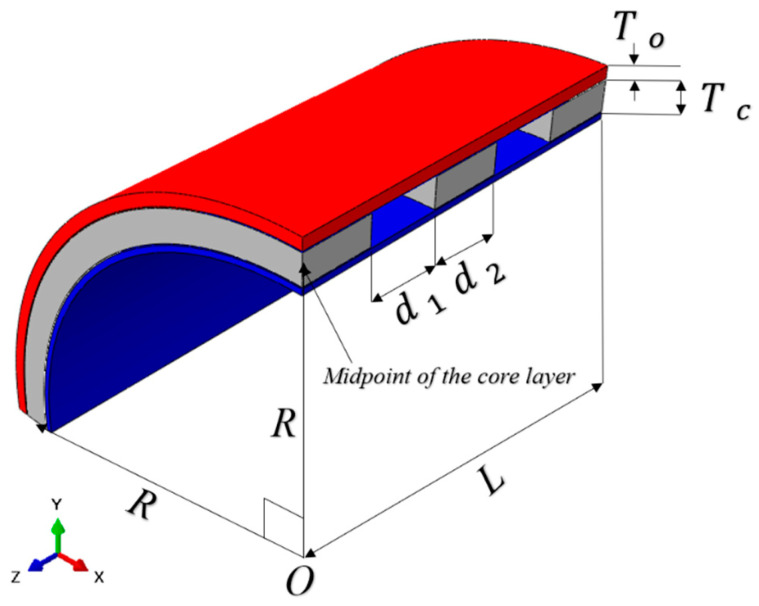
Typical structure schematic diagram.

**Figure 2 materials-17-00595-f002:**
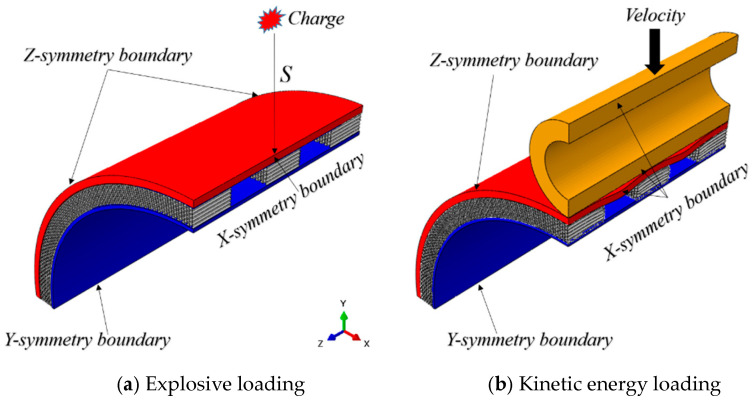
Double impact load on cylindrical structure diagram.

**Figure 3 materials-17-00595-f003:**
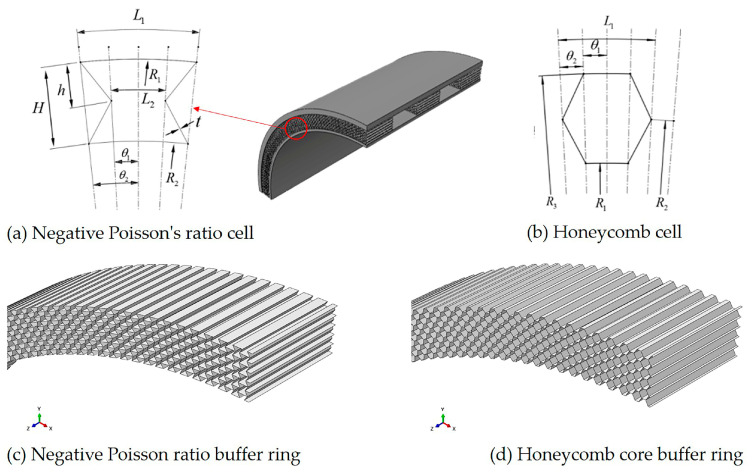
Buffer ring structure and cell size diagram.

**Figure 4 materials-17-00595-f004:**
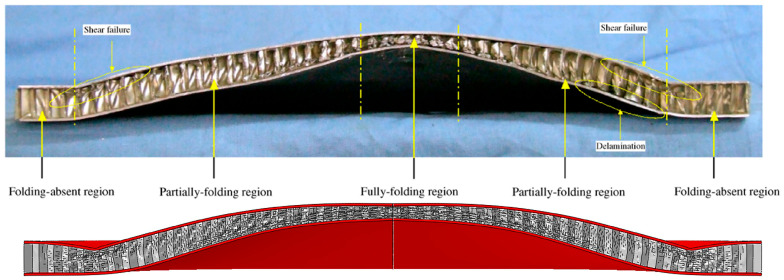
A comparison of the deformation and failure patterns of the honeycomb sandwich structure of experiment results and numerical prediction.

**Figure 5 materials-17-00595-f005:**
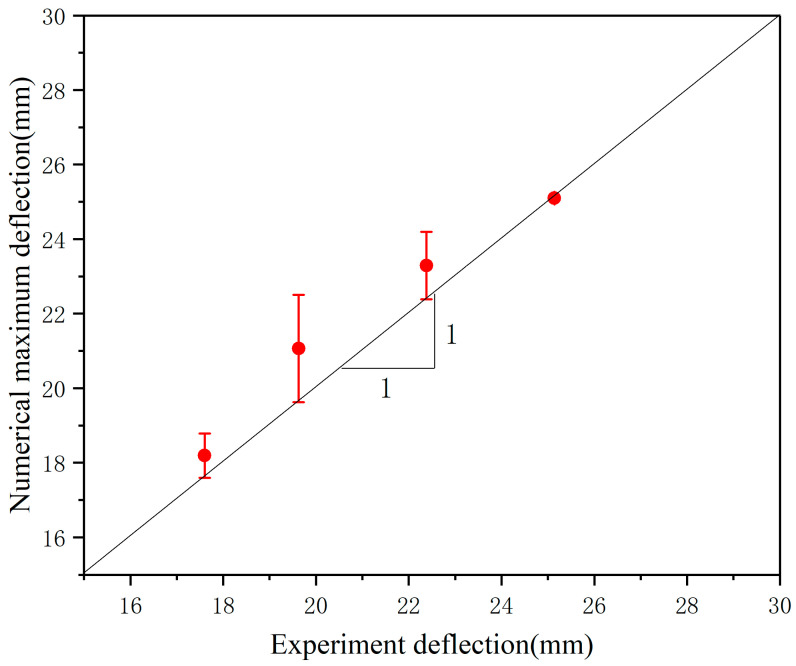
Comparison between the experimental and numerical predicted maximum central deflection of honeycomb panel.

**Figure 6 materials-17-00595-f006:**
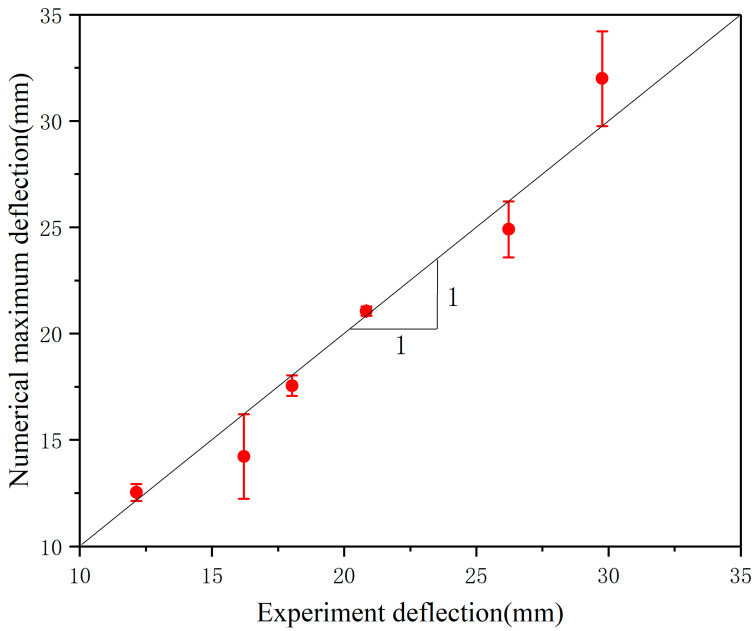
Comparison between the experimental and numerical predicted maximum central deflection of foam panel.

**Figure 7 materials-17-00595-f007:**
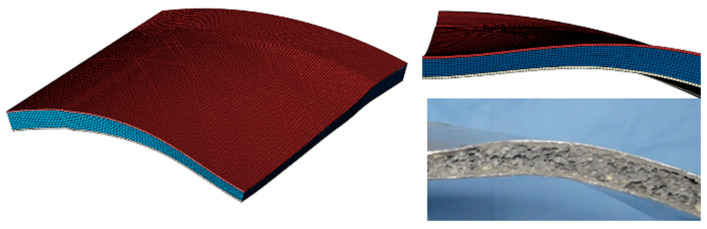
A comparison of the deformation and failure patterns of the foamed aluminium structure of experiment results and numerical prediction.

**Figure 8 materials-17-00595-f008:**
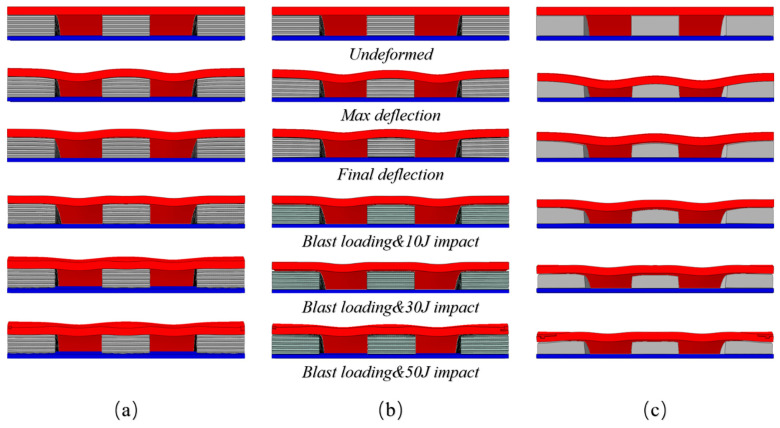
Comparison of three kinds of structural deformation of G1_3_A/H/F specimen ((**a**); AUX, (**b**): HEX, (**c**): FOAM).

**Figure 9 materials-17-00595-f009:**
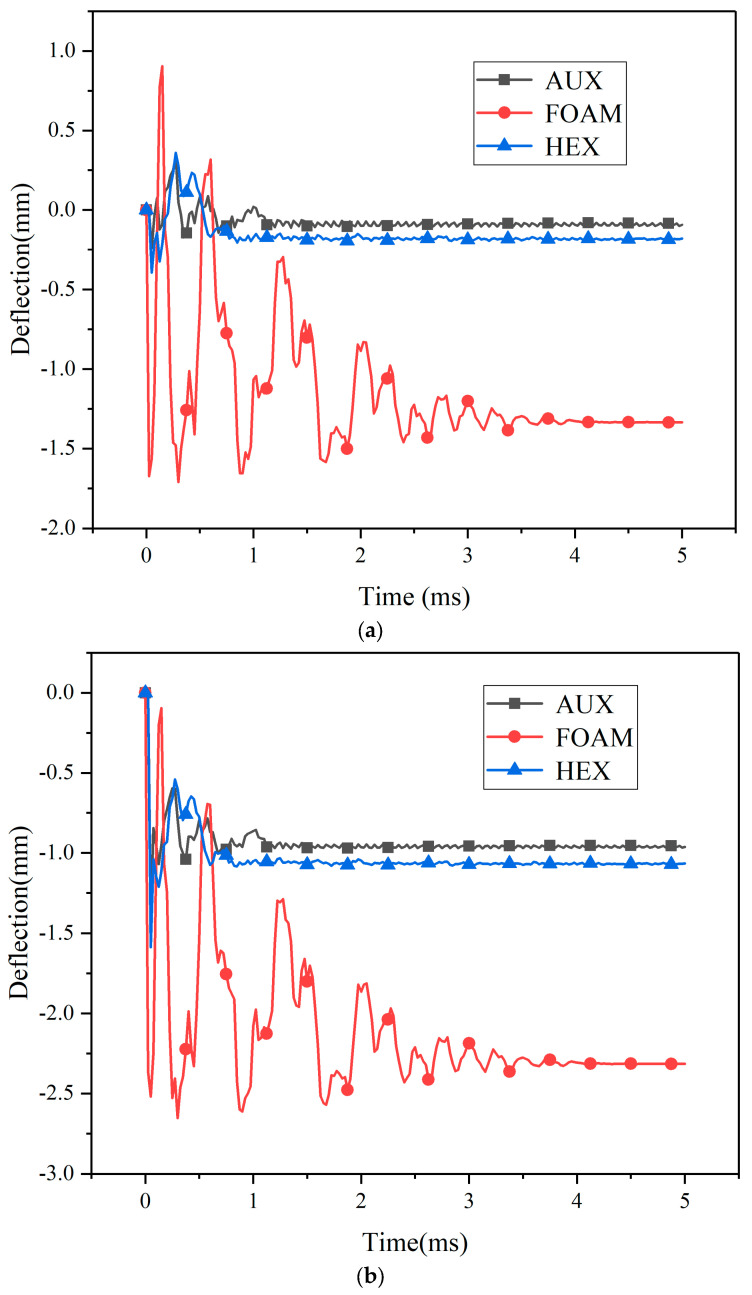
Specimen displacement—time history of blast loading ((**a**): central point (**b**): maximum displacement point). (**a**) Specimen central point displacement—time history of blast loading. (**b**) Specimen maximum displacement point displacement—time history of blast loading.

**Figure 10 materials-17-00595-f010:**
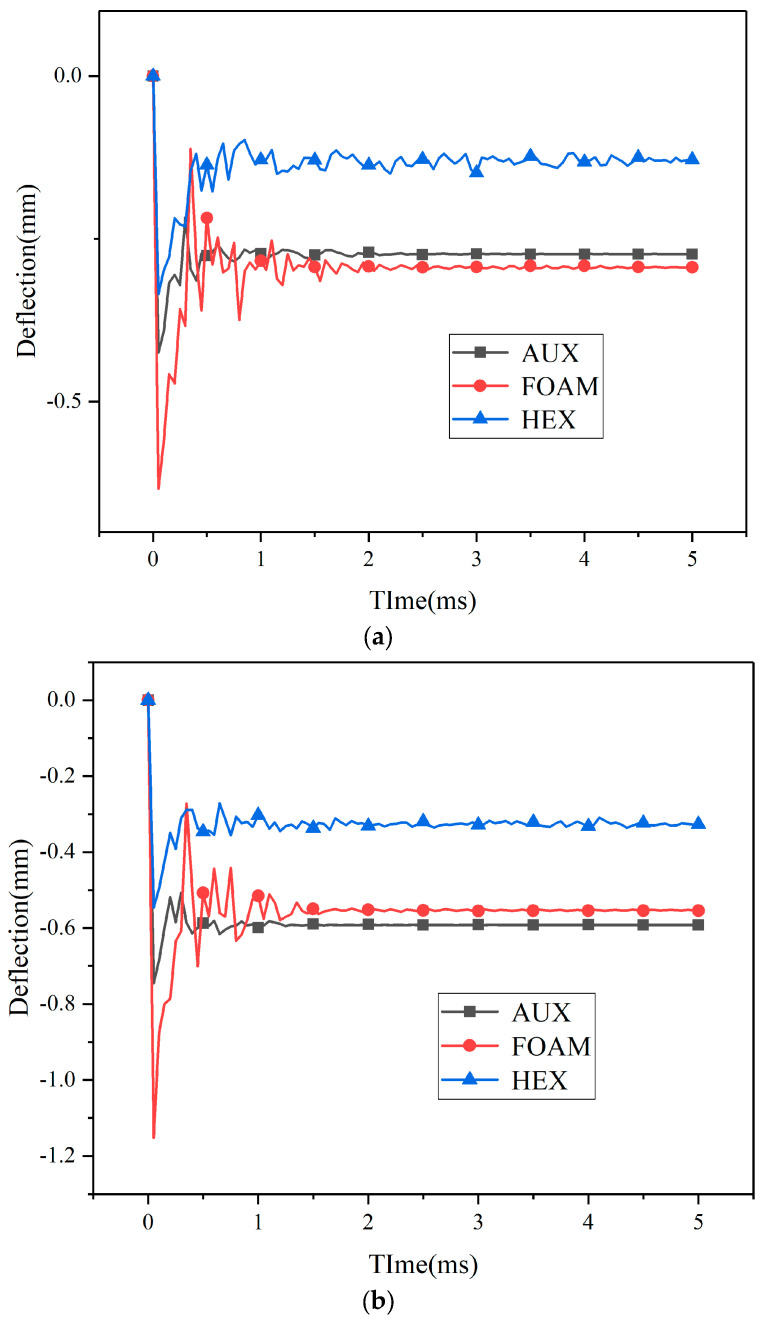
Center point displacement of kinetic energy impact loading specimen—time history ((**a**): 10 J, (**b**): 30 J, (**c**): 50 J). (**a**) Center point displacement of 10 J kinetic energy impact loading specimen—time history. (**b**) Center point displacement of 30 J kinetic energy impact loading specimen—time history. (**c**) Center point displacement of 50 J kinetic energy impact loading specimen—time history.

**Figure 11 materials-17-00595-f011:**
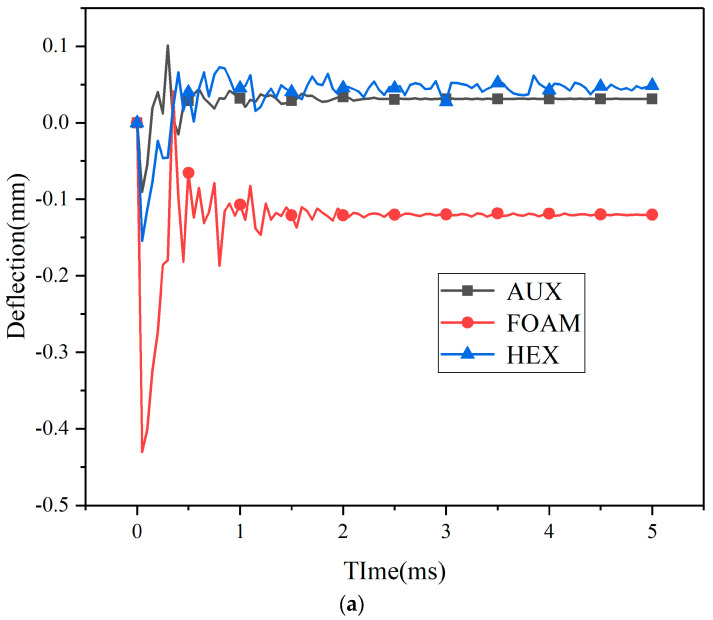
Maximum displacement point displacement of kinetic energy impact loading specimen—time history ((**a**): 10 J, (**b**): 30 J, (**c**): 50 J). (**a**) Maximum displacement point displacement of 10 J kinetic energy impact loading specimen—time history. (**b**) Maximum displacement point displacement of 30 J kinetic energy impact loading specimen—time history. (**c**) Maximum displacement point displacement of 50 J kinetic energy impact loading specimen—time history.

**Figure 12 materials-17-00595-f012:**
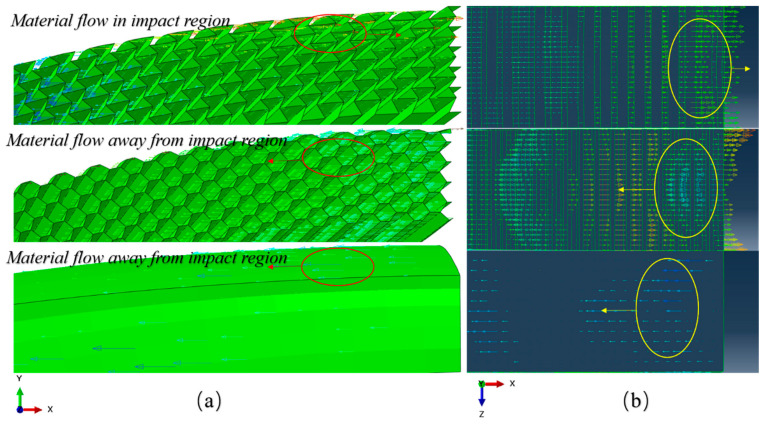
Core layer material velocity vector graph ((**a**): blast loading, (**b**): kinetic energy impact loading).

**Figure 13 materials-17-00595-f013:**
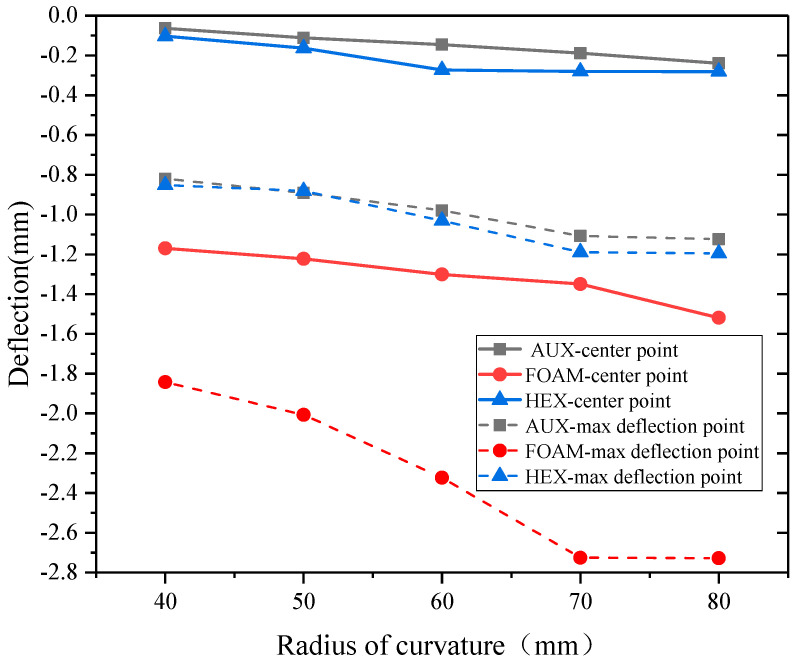
Displacement—curvature radius relation under blast loading.

**Figure 14 materials-17-00595-f014:**
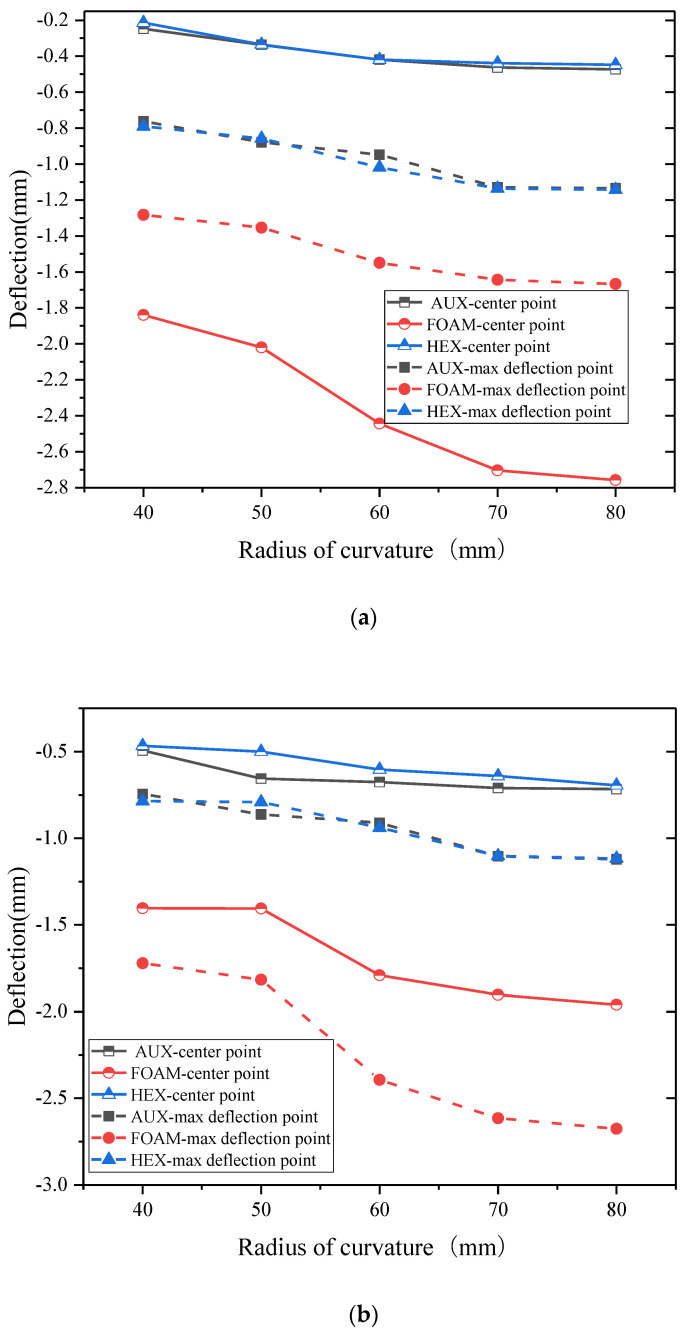
Displacement—curvature radius relation under double impact loading((**a**): 10 J, (**b**): 30 J, (**c**): 50 J). (**a**) Displacement—curvature radius relation under blast loading and 10 J kinetic energy impact loading. (**b**) Displacement—curvature radius relation under blast loading and 30 J kinetic energy impact loading. (**c**) Displacement—curvature radius relation under blast loading and 50 J kinetic energy impact loading.

**Figure 15 materials-17-00595-f015:**
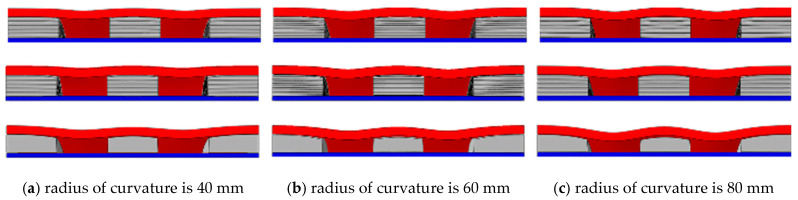
The contrast of three structural deformations of different curvature radii under blast loading ((**a**): 40 mm, (**b**): 60 mm, and (**c**): 80 mm).

**Figure 16 materials-17-00595-f016:**
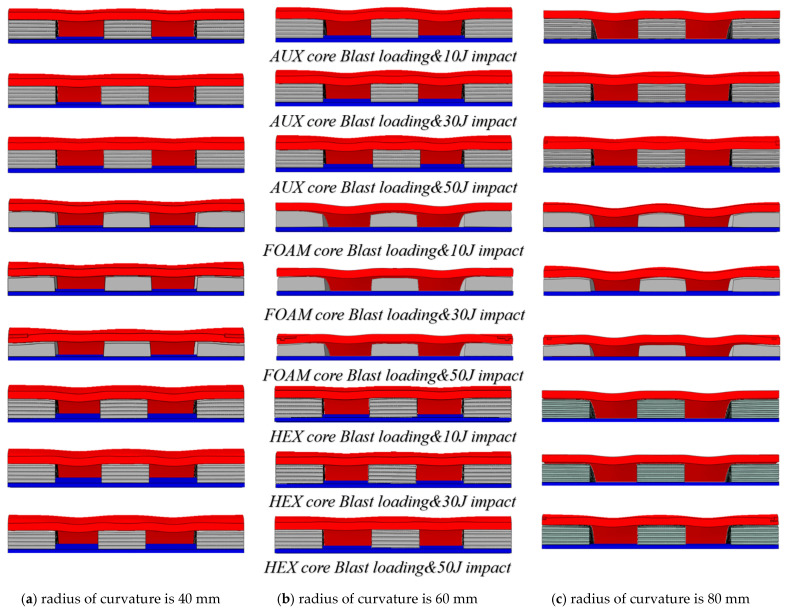
The contrast of three structural deformations of different curvature radii under double impact loading ((**a**): 40 mm, (**b**): 60 mm, and (**c**): 80 mm).

**Figure 17 materials-17-00595-f017:**
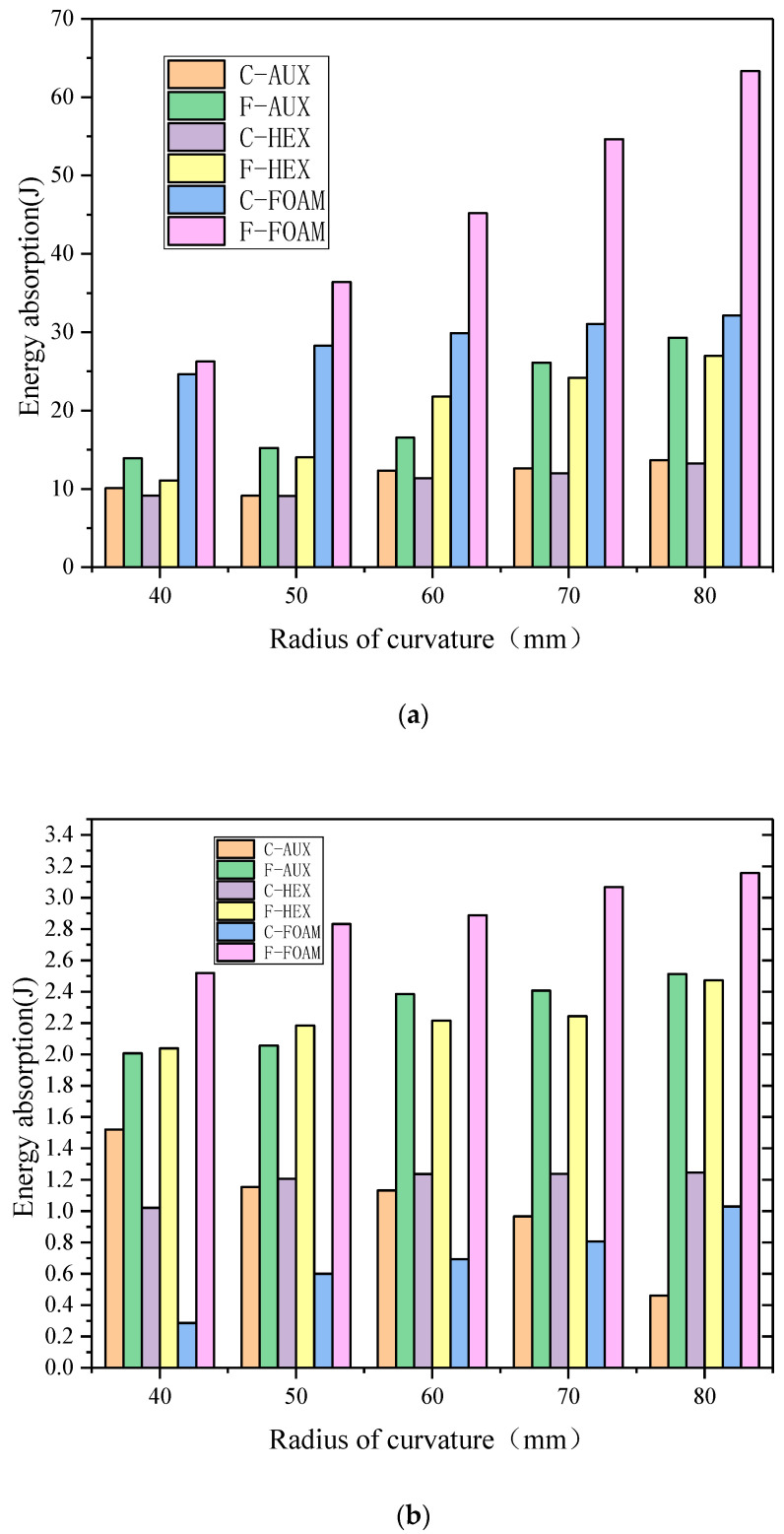
Energy absorption of each part of the three structures under different curvature radius ((**a**): blast loading, (**b**): 10 J impact, (**c**): 30 J impact, (**d**): 50 J impact).

**Figure 18 materials-17-00595-f018:**
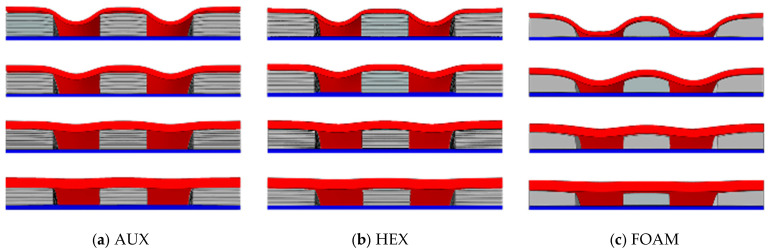
Comparison of deformation of three structures with different thickness combinations ((**a**); AUX, (**b**): HEX, and (**c**): FOAM).

**Figure 19 materials-17-00595-f019:**
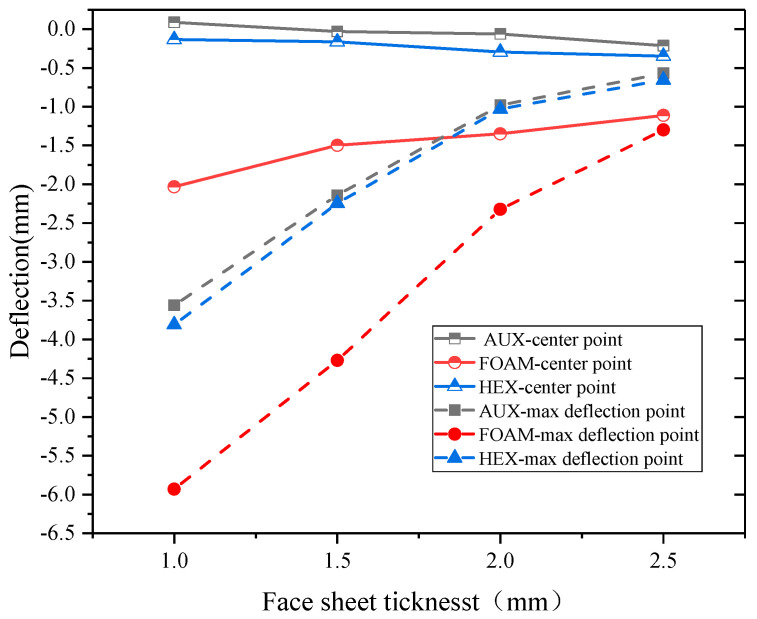
Displacement-panel thickness relation under blast loading.

**Figure 20 materials-17-00595-f020:**
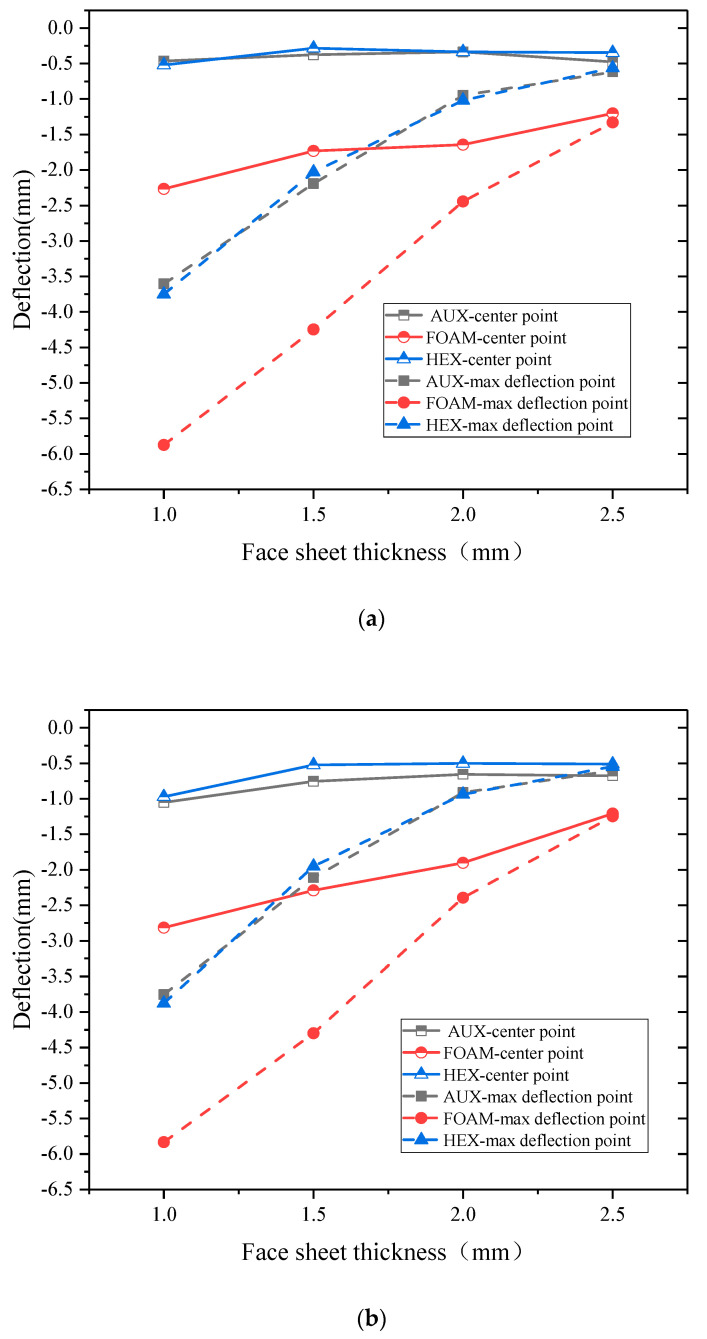
Displacement—panel thickness relation under double impact loading ((**a**): 10 J, (**b**): 30 J, (**c**): 50 J). (**a**) Displacement—panel thickness relation under blast loading and 10 J kinetic energy impact loading. (**b**) Displacement—panel thickness relation under blast loading and 30 J kinetic energy impact loading. (**c**) Displacement—panel thickness relation under blast loading and 50 J kinetic energy impact loading.

**Figure 21 materials-17-00595-f021:**
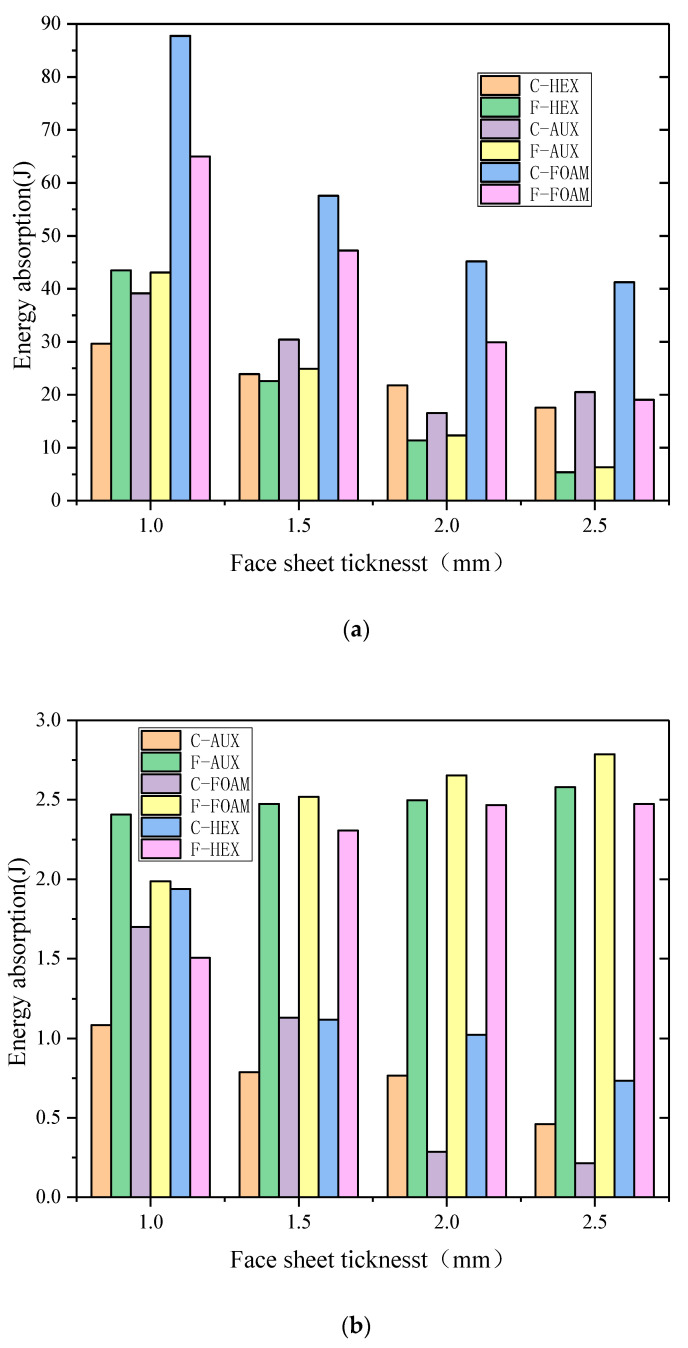
Energy absorption of each part of the three structures under different thickness combinations ((**a**): blast loading, (**b**):10 J impact, (**c**): 30 J impact, and (**d**): 50 J impact).

**Figure 22 materials-17-00595-f022:**
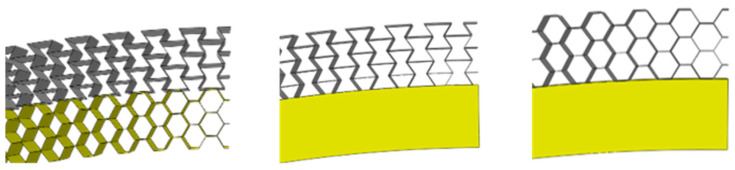
Structure diagram of composite buffer ring.

**Figure 23 materials-17-00595-f023:**
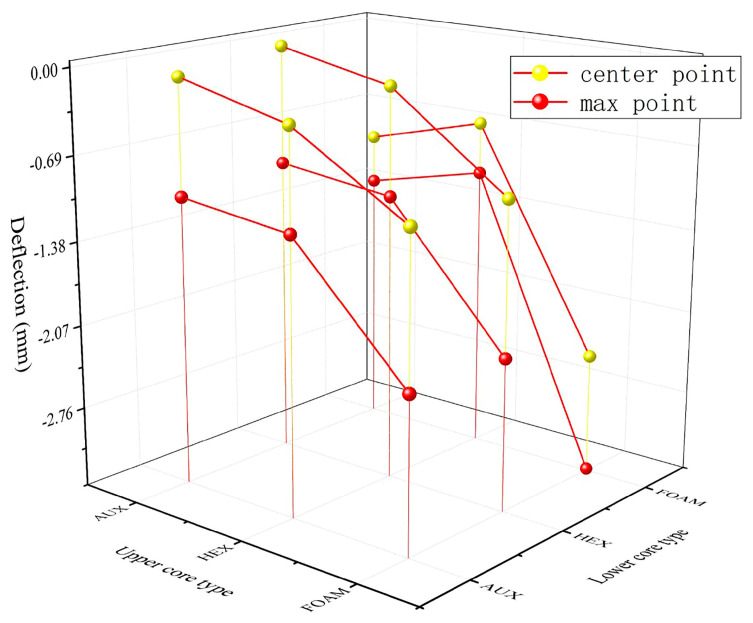
Displacement-combination mode relation under blast loading.

**Figure 24 materials-17-00595-f024:**
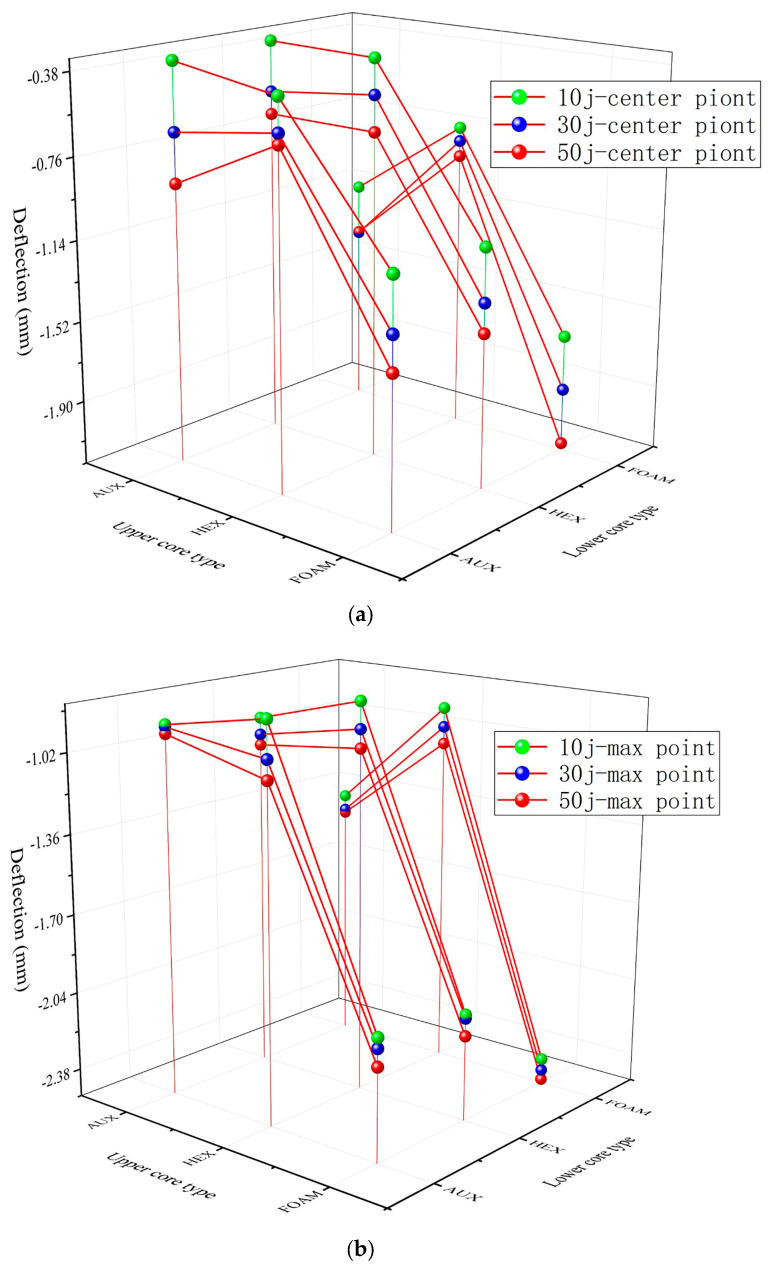
Displacement-combination mode relation under double impact loading (a: central point b: maximum displacement point). (**a**) Central point displacement-combination mode relation under double impact loading. (**b**) Maximum displacement point displacement-combination mode relation under double impact loading.

**Figure 25 materials-17-00595-f025:**
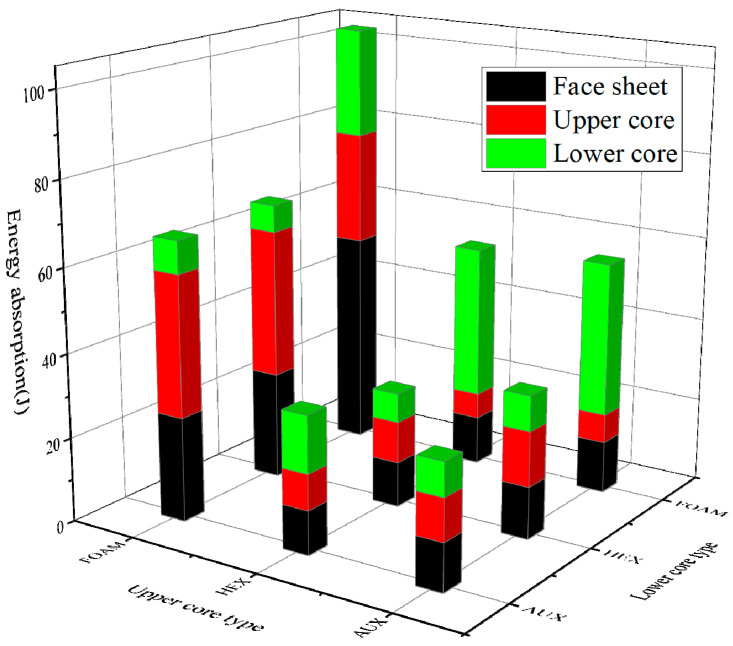
Energy absorption of each divided structure in different combinations under blast loading.

**Table 1 materials-17-00595-t001:** Specimen geometry parameter table.

Parameter Name	Radius of Curvature	Axial Length	Panel Thickness	Core Thickness	Core Interval	Core Width
Parameter value (mm)	40, 50, 60, 70, 80	60	1, 1.5, 2.5, 2	4.5, 5, 5.5, 6	12	12

**Table 2 materials-17-00595-t002:** Aluminum alloy material parameter table [8].

Material	Density (kg/m^3^)	Young’s Modulus (GPa)	Yield Stress (MPa)	Poisson’s Ratio	Tangent Modulus (GPa)	α	β
Al-2024	2680	72	75.8	0.33	0.737	1.76	1.44 × 10^−4^
Al-3104-H19	2720	69	262	0.34	0.69	2.63	2.17 × 10^−4^

**Table 3 materials-17-00595-t003:** Material constants for aluminum foam [9,25].

	$\sigma_{p}\left(MPa\right)$	$\alpha_{2}\left(MPa\right)$	1/β	$\gamma \left(MPa\right)$	$E_{p}\left(MPa\right)$
$C_{0}\left(MPa\right)$	0	0	0.22	0	0
$C_{1}\left(MPa\right)$	590	140	320	42	3.3 × 10^5^
k	2.21	0.45	4.66	1.42	2.45

**Table 4 materials-17-00595-t004:** Standard kinetic body geometry parameters.

Inside Diameter (mm)	Outside Diameter (mm)	Virtual Density (kg/m^3^)	Quality (kg)
15	25	53.05	1

## Data Availability

Data are contained within the article.

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
