# Peer review of "A Comparative Study on Impact Resistance of Cylindrical Structures with Cushioning Energy Absorbing Rings under Double Impact Loading"

_materials, 2024, doi:10.3390/ma17030595_

Round 1

Reviewer 1 Report

Comments and Suggestions for Authors

Since all the comments / suggestions provided by the reviewers are addressed satisfactorily, the manuscript can be considered.

Author Response

The article has been modified according to the reviewer's comments

Reviewer 2 Report

Comments and Suggestions for Authors

This manuscript examines the impact resistance of circular cylindrical sandwich structures under different loading conditions. Two types of aluminum cell structures, as well as aluminum foam, were considered as core materials for the cylinders. The impact loadings were due to explosion shock waves followed by direct impact loading from a cylindrical impactor onto the outer surface of the specimen. I believe the authors should address the following concerns:

1. There are numerous typos, grammatical errors, and journal style problems in the manuscript. Some citation numbers are incorrectly placed. Additionally, there are inaccuracies in the names of journal authors. For instance, in papers with multiple authors, it's appropriate to use the first author's name followed by 'et al.' Furthermore, 'J' in joule and 'P' in Pascal should be capitalized. Spaces are needed before parentheses, and so on.

2. The figure captions are incomplete; they should provide explanations for all sections ((a), (b), ...) of the figures.

3.  As far as I know, there are numerous published works exploring the axial and radial impact resistance of cylinders that have not been cited in the manuscript.

4. Line 38: what is the meaning of “with the tensile behavior” for materials with negative Poisson’s ratio.

5. The authors did not explain the real application of the two-step impact that they defined.

6.   The relative density of the buffer rings was assumed to be 0.148. However, within the paper, it's important to elaborate on the rationale behind assuming a range for the shell thickness concerning AUX and HEX configurations.

7. The strength of the core-to-face sheets connection significantly influences the structure's impact resistance. However, the manuscript lacks an explanation of the adhesive employed. How realistic is the suggestion that the core would not separate from the face sheets during impact?

8.  It seems that Eqs. (1) through (5) in the manuscript were derived using the methodology outlined in Ref. [15]. It's crucial to specify the citations for equations clearly to attribute the methodology or formulas to the appropriate source.

9. Line 99: Is there a standard for the impact of two cylinders? If yes, specify the standard number; if no, justify the reason for selecting this size and material specification for the impactor.

10. Line 211: Define “ACG-1/4-TK-5”.

11. Line 232: Define “sxl=250x250mm”.

12. The title of the vertical axis in Figs. 5 and 6 is incorrect (i.e., 'Numerical prediction (mm)'). It should be revised to something like 'Numerical maximum deflection (mm)' for accurate representation.

13. Table 4 appears messy, lacking any discussion. If the results featured in your subsequent graphs are derived from this table, redundant repetitions should be avoided. Additionally, the designation 'A/H/F' is redundant in all specimen names. It would be more appropriate to remove this designation and provide an explanation in the table title.

14. Line 262: “Under the effect of explosive load, the deformation of the panel center and the maximum deformation point is the same.” Why have you presented different results for the deformation of the panel center and the maximum deformation point for this section?

15. The horizontal axis in Figs. 9-11 needs correction (Time (ms)). These figures depict specimens with damping. Could you specify where the damping is accounted for in your material models?

16.  Provide details on how you calculate the absorbed energy during impact. Additionally, explain the method used to compute the portion of energy absorbed by the face sheet and the core separately.

17. Line 332: Plastic deformation absorbs the impact energy, while elastic deformation does not absorb or dissipate energy.

18. Figs. 8, 15, 16, and 18 do not provide a clear comparison between different structures.

19. Section 5: The first paragraph on page 31 is unclear. Were two core layers used in the sandwich cylinder? If so, it would be beneficial to include a schematic figure to illustrate this configuration for better understanding.

20. Your results indicate that the cylinder with a foam core absorbs more energy. However, your conclusion seems to differ from this observation.

Author Response

The article has been modified according to the reviewer's comments,the details are as follows

1. The errors and grammar in the manuscript have been modified, and the article has been modified according to the style of the journal, and the corresponding units and formats have also been modified

2. Explanations are added to each picture separately

3. The introduction part was rewritten and the references were added accordingly

4. It is explained here that the structure with negative Poisson's ratio has the phenomenon of tension, that is, in the case of transverse stretching, the longitudinal will not compress but expand

5. Two-step influence refers to the fact that in practical applications, the load of explosion shock wave will take precedence over the performance of the specimen under the action of kinetic energy load, and the kinetic energy load will reach the surface of the specimen after its full action. The multi-step method can effectively reduce the complexity of the simulation model, save computing resources, and improve calculation accuracy

6、In this paper, three different buffer ring structures are set as the same relative density of 0.148.  In order to ensure the same relative density of the three buffer ring structures, the method of controlling the thickness of the shell element is adopted in this paper.  The width of the buffer ring structure is the same, and the total length of the side cell of the buffer ring structure is different by integrating the total length of the side cell of the different sibling structure.  By adjusting the thickness of the shell ele-ment, the product of the three is the same.  For the same cell material, that is, the total mass of the buffer ring structure, the relative density of the buffer ring structure is the same. Changes have been made in the manuscript.

7. In section 3.4.1 of the paper, it can be seen from the photos of the experimental results and the results of simulation analysis that there is no separation between the core layer and the panel. If the adhesive unit is used, or the calculation amount of simulation calculation is increased, it is not explained in detail in the manuscript

8. The formula is derived from Abaqus analysis user's manual, and citations and references have been added

9. The choice of size is based on the specific structural size of a certain type of weapon equipment, and the size of the specific protective structure and the structural size of the cell are determined according to this equipment

10. The code name ACG-1/4-TK-5 comes from the references cited in the paper, which represents the fifth group of experimental specimens with honeycomb sandwich plates

11. The expression of sxl=250x250mm comes from the references cited in the paper, that is, the length times width of the area receiving the impact is 250mm times 250mm

12. Figure 5 and Figure 6 have been modified according to the reviewer's comments

13. Table 4 has been attached to the schedule

14. It is said here that the deformation process of both is basically the same and has been corrected in the manuscript

15. The horizontal coordinate in Figure 9-11 has been corrected. As for the vibration process of deformation, it is mainly the elastic deformation of the front panel, which leads to the shock of the front panel. At the same time, the whole core structure also has elastic deformation, resulting in vibration

16. The energy absorption calculation mainly comes from the calculation of total strain energy in ABAQUS software, and the energy absorption is calculated by integrating the energy of each node unit; The energy absorbed by the panel and the core layer is calculated as the energy integration of the panel and core element nodes respectively

17、It was misexpressed earlier,the whole structure can react to shocks, so the structure undergoes greater elastic deformation, producing more oscillations, and dissipating energy.Changes have been made in the manuscript.

18. For the comparison of deformation results of different parameters, the transverse comparison can clearly compare the deformation of different structures

19. New schematics have been added

20, the specimen with the foam core will indeed absorb more energy, but at the same time, it will also produce a large shape deformation, and its energy absorption density is less than the other two core layers, so it shows poor impact resistance

Reviewer 3 Report

Comments and Suggestions for Authors

After checking the manuscript, my observations / comments to be transmitted to the authors are the following ones:

-e-mail address must not be provided in between brackets along with the name of the authors

-details regarding affiliation are incomplete

-Abstract section should be reconsidered – avoiding general information and expressions like “The research results can provide reference....” Can or it really provides???? Information about the objective of the results, methodology followed and reached results have to be provided much clearly in the Abstract section in my opinion.

-The methodology emphasized in the paper is quite well explained in terms of design, realized analyses and tests that have been performed. In section 2 information on how the patterns of the honeycomb sandwich structure shown in Figure 4 have been produced should be clearly described.

-I have found the article to be consistent in terms of scientific content. In terms of lenght of the manuscript – 37 pages are not so easy to be read and followed – is a way to edit this manuscript and make it little but shorter (like the article to have 22-25 pages maximum maybe)? Some photos or tables can be provided in appendices to this paper I think!

-One scheme with testing procedure of structures shown in Figure 8 can be included (this is a suggestion) in section 2 along with some explanations. What is given in Figure 8-11, Figure 14-17, Figure 18-21 are quite similar from the approaching point of view (even the parameter analyzed are different). Is there a possibility to combine all these in a single section, but in much syntehtic way?

-Maybe this section can end up with one single set of conclusions and some discussions in terms of reached results to the ones provided in the literature for a simplier way of following the flowing of the article and its logics.

-Section 5 can be much better explained in the paper. Which is the main purpose of this section – to finalize optimize some of the parameters? There have been used some data in this sense taken from section 4? (it is not clear).How have been used the results reached in Figure 24 in the experiments is not at all clear! One section of discussions to interpret the data in the light of finding of other researchers that have been doing research in the field should be introduced before the Conclusions section (number 6) here.

-Section 6 (Conclusions) should be reconsidered – it has to be more than a repetition of things that have been already presented inthe paper. What is still necessary to be done in the future in this direction and how – this is not clear! Are there any aspects still necessary to be improved?Which are these aspects and how can the research be continued?

-References section has to be reconsidered as well. References are not provided in accordance with the requirements of MDPI Materials journal template. 4 articles dated from 2022 (out of 20), zero from 2023...most of references quite old – I personally suggest that the references section should be consistently updated in order to have much updated resources referred in the paper and more than 20. I suggest that at least 30 references have to be referenced in the paper, out of which (this is my recommendation) at least half of these references are dated in 2022 / 2023.

I have checked the article with Ithenticate application – and the similiarity index is too high (please see the attached report generated with Ithenticate application after checking / reviewing the paper)!

-Similarity index is referring to the article entitled „A Comparative Study on Impact Resistance of Composite Buffer Ring Structures Under Double Impact Loading” publsihed by Bo Zhang, Shunshan Feng and Lin Chen”  in the „Material Strength and Applied Mechanics” – IOS e-book – see: https://ebooks.iospress.nl/doi/10.3233/ATDE39

The authors are encouraged to provide original manuscript for MDPI Materials journal. The similarity index is very very high in my opinion (on unacceptable rate) – Due to this reason, I was almost put in the position to reject this manuscript and suggest the authors to re-submit the manuscript with original elements included (and lower rate of similiarity index) but I have decided to go with major revisions required in my decision, notifying the editor about this issue which is the most important aspect to be improved in the next variant of manuscript before the article could be considered to be published in MDPI Materials journal.

Author Response

The article has been modified according to the reviewer's opinions. The following is the specific modification content

1. The email was deleted according to the reviewer's advice

2. Revised the details

3. The summary has been revised

4. This part has been rewritten in the article

5. The excessively long tables in the text have been placed in the attached tables to reduce the length of the manuscript

6, for the comparison of deformation results of different parameters, the horizontal comparison can clearly compare the deformation of different structures

7. The interior of the fifth section has been rewritten and a new diagram has been added to illustrate it

8. The conclusion section has been added

9. The reference literature has been re-planned

10. The paper with excessively high repetition rate is another paper published by me with some repetition in terms of writing, grammar and research background, which has been revised and the revised manuscript has been resubmitted.

Round 2

Reviewer 2 Report

Comments and Suggestions for Authors

The authors' response to the raised comments is poor. They should specify the changes in the manuscript and their locations. Additionally, the authors did not respond to Comment No. 15, which is related to damping in material models. It appears that this critical question should be addressed and explained in the manuscript, especially considering that their results indicate damping in the time histories of deformation of the structure.

Author Response

When uploading the last modified version, I have already highlighted the modified part in yellow. Maybe the website will automatically update it to the form of red font and underline, resulting in unclear visual effect.

In the whole simulation model, damping is divided into two parts, namely volume viscosity and material damping.

The volumetric viscosity is used to introduce damping due to volumetric strain, and is particularly necessary when studying higher-order performance in high-speed dynamic analysis. Volumetric viscosity is introduced only as a numerical effect, so the stress at the point of the material does not take into account the influence of volumetric viscous pressure. Specific parameter Settings have been added to the manuscript changes.

Material damping is Rayleigh damping, which contains two damping parameters: mass proportional damping is the proportional coefficient of the mass matrix, which is mainly used to eliminate low-order oscillations; Proportional damping of stiffness is the proportional coefficient of the stiffness matrix, which is mainly used to eliminate higher order oscillations. Specific parameter Settings have been added to the manuscript changes.

Reviewer 3 Report

Comments and Suggestions for Authors

The authors have managed to address the major concerns regarding this paper. I appreciated the reformulated aspects in the manuscript + new details added which in my opinion brings scientific quality to this paper.

I have been re-cheching the article with Ithenticate application and I was noticing that the similarity index has droped from 30 % to 16 % in the new variant.

Similarity index is high especially referring to aspects undertaken from the article entitled „A Comparative Study on Impact Resistance of Composite Buffer Ring Structures Under Double Impact Loading” published by Bo Zhang, Shunshan Feng and Lin Chen”  in the „Material Strength and Applied Mechanics” – IOS e-book – see: https://ebooks.iospress.nl/doi/10.3233/ATDE39

I am attaching the Ithenticate result (generated document) to this reviewing and is going to be the editor who will decide if 16 % similarity index is acceptable or there are few things recommended to be reformulated as they were undertaken from this paper: „A Comparative Study on Impact Resistance of Composite Buffer Ring Structures Under Double Impact Loading” publsihed by Bo Zhang, Shunshan Feng and Lin Chen”  in the „Material Strength and Applied Mechanics” – IOS e-book – see: https://ebooks.iospress.nl/doi/10.3233/ATDE39 in order to decrease the similarity index little bit more.

With this comment I will go therefore with my decision of accepting this paper after minor revisions still to be done (there are also some editing issues present in text needed to be sorted out so the paper will fully respect the MDPI Materials template requirements (see for example References section in the end.

Author Response

The manuscript was modified according to the reviewer's comments.The format of the manuscript has been modified, and the pictures marked by the reviewer have been changed.